# When are radiology reports useful for training medical image classifiers?

## Abstract

When exploring how radiology reports can be leveraged *during training*, prior works are limited to evaluating pre-trained image representations by fine-tuning to predict diagnostic labels, often extracted from reports, ignoring tasks with labels that are weakly associated with the text. To address this gap, we conduct a systematic study of how radiology reports can be used during both pre-training and fine-tuning, across diagnostic and prognostic tasks, and under varying training set sizes. Our findings reveal that: (1) Leveraging reports during pre-training is beneficial for downstream classification tasks where the label is well-represented in the text; however, image-text alignment can be detrimental in non-diagnostic settings where it's not; (2) Fine-tuning with reports can lead to significant improvements.

## 1 Introduction

Radiology reports containing key findings from medical images are routinely produced in clinical practice. These texts have recently drawn significant interest in machine learning research [1, 2, 3], and have been shown to be predictive of various patient outcomes, such as readmission [4] and ICU mortality [5]. Requiring manually written reports for test-time predictions is undesirable, but their prevalence in retrospective data makes them attractive to use during training. In particular, reports can be used in data-scarce settings to improve training of image classifiers, either by incorporating them i) in a *pre-training* objective [6], or ii) as privileged information (PI) when *fine-tuning* for a specific task [7]. Prior work has almost exclusively focused on the potential role of reports in *pre-training* image encoders—either incorporating them as supervision through multimodal objectives [8, 9], or omitting them in favor of self-supervised learning [10, 11]—with evaluation performed via report-free fine-tuning on *diagnostic* classification tasks, where labels are often extracted from the reports themselves in a rule-based manner [12, 13, 14]. As a result, the impact of report-based pre-training on tasks beyond *diagnosis*, and the potential utility of reports during *fine-tuning*, remain largely unexplored.

To address these gaps, we investigate the usefulness of radiology reports for training image classifiers during both pre-training and fine-tuning. We find that: (1) Pre-training with report supervision is beneficial for diagnostic tasks at moderate sample sizes when the label is *strongly* correlated with the text. (2) Explicitly aligning image and text embeddings hurts downstream performance when the label is not captured well by the report, something that is more prominent for non-diagnostic tasks. Crucially, methods relying on text supervision *in addition to* self-supervision avoid this pitfall. (3) Incorporating privileged reports during fine-tuning can yield substantial gains in accuracy.

## 2 Experiments

We compile 5 experiments (described further in Appendix A.1) from two existing datasets, MIMIC-CXR [12] and INSPECT [20]. In these, we extract features from images using 7 frozen backbones

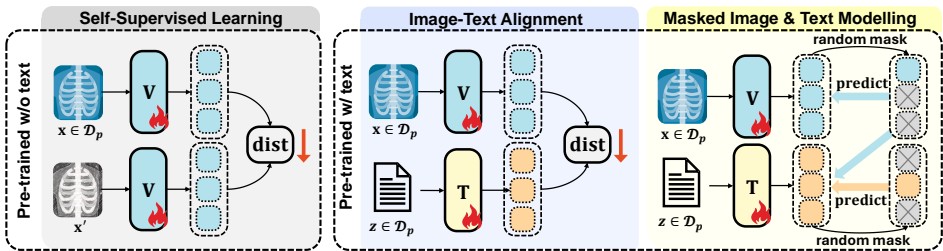

Figure 1: Three categories of pre-training objectives considered in this work. *Self-Supervised Learning:* RAD-DINO [10], Medical MAE [15], and C2L [16] all leverage a self-supervised learning (SSL) objective without report supervision, which learns an image representation invariant to random augmentations. *Image-Text Alignment:* BioViL-T [17], GLoRIA [8] and BiomedCLIP [18] use text supervision in a CLIP-style setup [19] by aligning image and text representation. BiomedCLIP has been trained on more general biomedical images; hence, radiographs and medical reports constitute a far smaller portion of its training set. *Masked Image & Text Modelling:* Lastly, we include MRM [9], which is pre-trained with SSL and report supervision, but without explicit multi-modal alignment.

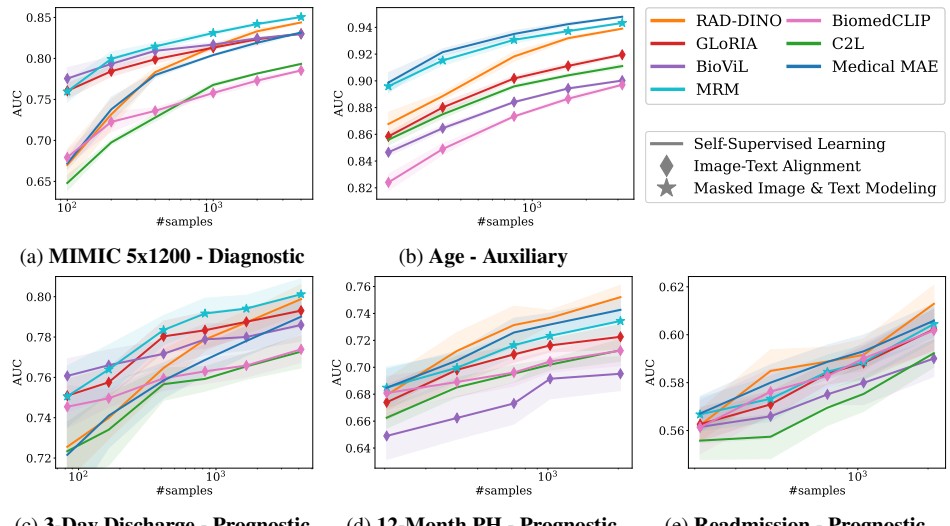

(a) **MIMIC 5x1200 - Diagnostic**   (b) **Age - Auxiliary**

(c) **3-Day Discharge - Prognostic**   (d) **12-Month PH - Prognostic**   (e) **Readmission - Prognostic**

Figure 2: Sample efficiency across 5 seeds (10 for **Readmission**). 90% CI shown as shaded area.

from 3 groups of pre-training methods, illustrated in Figure 1. When fine-tuning *with radiology reports*, we adopt generalized distillation [21], where an image-only student model distills from a teacher model that additionally has access to the corresponding report. We compare our results against self-distillation, where the teacher is simply another image-only model of the same type. Implementation and training details are provided in Appendix A.

## 2.1 Impact of pre-training with reports

**Explicitly aligning with reports limits generalizability.** As expected, we observe the highest relative performance of GLoRIA and BioViL compared to other models in the **MIMIC 5x1200** experiment (Figure 2a), where the labels have been extracted from the reports. This is mainly evident in the low sample regime, and RAD-DINO overtakes both when the number of samples increases to over $1,000$. However, when predicting **Age** (still using images from the MIMIC dataset), the relative performance of these algorithms decreases significantly compared to backbones pre-trained with self-supervision. Instead, Medical MAE, MRM, and RAD-DINO perform the best, with the previously poorly performing C2L achieving a higher AUC than BioViL, and only slightly lower than GLoRIA. The high sample efficiency of MRM in both experiments highlights the benefit of using text supervision as a complement to image self-supervision.

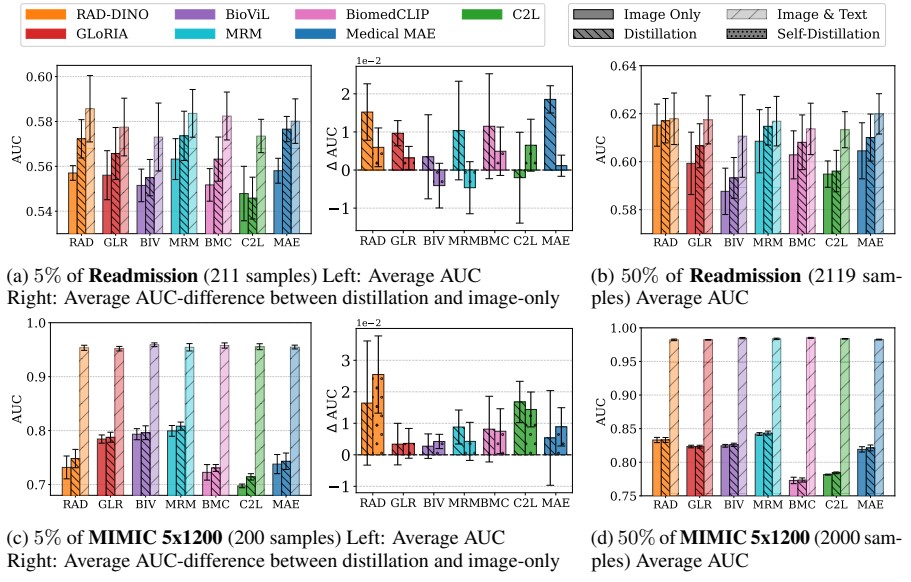

(a) 5% of **Readmission** (211 samples) Left: Average AUC Right: Average AUC-difference between distillation and image-only

(b) 50% of **Readmission** (2119 samples) Average AUC

(c) 5% of **MIMIC 5x1200** (200 samples) Left: Average AUC Right: Average AUC-difference between distillation and image-only

(d) 50% of **MIMIC 5x1200** (2000 samples) Average AUC

Figure 3: Distillation results averaged over 5 seeds. Error bars represent the $95\%$ CI.

**Image self-supervision is usually beneficial for prognostic tasks.** For the prognostic tasks, the order of performance varies. In **3-Day Discharge** (Figure 2c), GLoRIA, BioViL, and MRM (all pre-trained with text supervision) perform well in the low sample regime. Yet, in the **12-Month PH** experiment (Figure 2d), GLoRIA and BioViL do not see the same benefits, with BioViL performing the worst in all sample sizes. The trend is similar in **Readmission** (Figure 2e). We hypothesize that while these tasks are prognostic, whether or not a patient is discharged in the coming days may more closely correlate with a diagnostic label discussed in the reports. For example, it is possible for pneumothorax (a report-extracted label in MIMIC-CXR [12]) to persist for more than 3 days [22].

## 2.2 Fine-tuning with reports

**Distillation can have a larger impact than pre-training.** Interestingly, pre-training does not always have the largest effect on performance. In Figure 3a, distillation has a higher impact on the AUC than the choice of pre-training method when predicting **Readmission** in the small-sample domain. The results show that distillation from a teacher with access to the text report consistently increases the AUC across seeds, something that is not observed when applying self-distillation from an image-only model. As the number of samples grows (Figure 3b), distillation still leads to a meaningful performance increase, but the gap between different image backbones widens.

**Distillation performs poorly in the diagnostic setting.** On the other hand, the results of distillation on the **MIMIC 5x1200** dataset (Figures 3(c–d)) highlight that a strong multi-modal teacher does not imply that distillation will be beneficial. The student sees no more benefit in distilling from the teacher as opposed to performing self-distillation (Figure 3c). This aligns with previous research, which has observed that utilizing PI through distillation performs poorly if the information is too predictive of the label [23, 24]. The results indicate that this is true for diagnostic labels of this kind, further underscoring that the benefit from using reports for pre-training as opposed to generalized distillation depends on the task structure. Still, the high performance of models pre-trained with text in Figure 2a raises the question whether an alternate fine-tuning method might better leverage reports.

**Benefits of distillation are dependent on both task and backbone.** We don't always benefit from applying distillation, even in the prognostic setting. Beyond variance with the task, our results show that not all encoders benefit the same when fine-tuning with PI. Medical MAE, RAD-DINO, MRM and GLoRIA see large performance increases in the **Readmission** experiment (Figure 3a), while BioViL and C2L do not improve. The benefits are expected to depend on the relation between image $(X)$, text $(Z)$, and label $(Y)$. In practice, we have both the image and text backbones frozen, meaning that the performance will depend on the relation between the extracted features $V(x)$ and $T(z)$.

## 3 Potential Negative Societal Impact

Including texts during any part of training means including additional sensitive patient data. If the reports are not handled sensibly, with proper anonymization, patient profiling may be possible. Furthermore, while we believe prognostic predictions have great potential and should continue to be explored, we do not encourage blind implementation of such models. These labels can be subject to problematic correlations, raising additional bias and fairness concerns. For example, a person's wealth can affect whether they are readmitted or the type of treatment they can afford, and an image model could potentially pick up on proxies for this, such as ethnicity. This problem could be exacerbated for a teacher model that additionally has access to notes taken about the patient. Any practical adoption needs to explore these aspects, be aware of the limitations, and be employed accordingly.

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

## Appendix

This appendix includes additional experimental results and information about the training setup. Further details about data processing, backbone model usage, and hyperparameters are presented in Section A. Section B contains the results from distillation experiments on **Age** and **12-Month PH**, and an additional diagnostic experiment on the INSPECT dataset that reaffirms the limited benefit of distillation when reports are too predictive. In Section C we perform multiple ablation experiments that motivate hyperparameter choices, while showcasing the robustness of our results. Sections D and E cover compute usage and dataset licenses, respectively.

## A   Training & Data Processing

**Fine-tuning with reports**   When fine-tuning *with reports*, we adopt generalized distillation for privileged information (reports), as popularized by [21]. In this two-stage approach, we first train a teacher model $g$ using a cross-entropy loss $\mathcal{L}$ that, in addition to the image features $V(x)$, also has access to the text features $T(z)$, where $z$ is the accompanying radiology report and $T$ is a pre-trained text encoder (in our case a BERT model trained alongside BioViL-T [17]). In the second stage, we train a student model $f$ by distilling from the teacher $g$ according to

$$\min_f \sum_{i=1}^{N} \Big[ (1 - \lambda)\mathcal{L}\Big( f\big( V(\mathbf{x}_i)\big), y_i\Big) + \lambda \mathcal{L}\Big( f\big(V(\mathbf{x}_i)\big), g^\tau\big( V(\mathbf{x}_i), T(z_i)\big)\Big)\Big], \qquad (1)$$

where $\lambda \in [0, 1]$ is the imitation parameter that balances the importance of following the teacher prediction, and $g^\tau(\cdot)$ denotes the teacher prediction computed with a softmax temperature of $\tau > 0$.

**Fine-tuning architecture.**   We design $f$ to first perform self-attention on the visual tokens output from $V$, where the query, key and value networks are learnable linear heads. Then we perform mean pooling on the self-attention tokens output and feed the resulting feature into a learnable linear classifier. The teacher network $g$ processes the text tokens output from $T$ with a separate self-attention and mean pooling step. Then, the pooled visual and text features are concatenated as the input to a single linear classifier. We choose this design as it performs well empirically (comparisons in Appendix C), without over-complicating the fine-tuning stage.

**Training Details**   All models were trained with a learning rate of $10^{-4}$ using the Adam optimizer and a batch size of 64. Performance was evaluated by measuring AUC after every epoch, with one-versus-rest and micro averaging for the multiclass problems. When training the teacher model, we save the version with the highest AUC on the validation set. We used the distillation temperature parameter $\tau = 0.25$ for all datasets except MIMIC 5x1200, which used $\tau = 2.5$. Experiments are averaged over multiple seeds with different network parameter initializations and training subsets.

When processing radiology reports in MIMIC-CXR we extract the impressions section and, if available, the findings section. INSPECT is a multimodal dataset containing CT images and pre-extracted impression sections from their accompanying reports. We perform digital radiograph reconstruction using the Plastimatch software suite [25] to convert the CT volumes to (anterior-posterior) radiographs that our pre-trained backbones can process. After extracting the radiograph, we follow the preprocessing of [12] and apply histogram equalization using OpenCV [26], before storing the images in the JPEG format with a 95 quality factor. The code used to process the INSPECT volumes will be made available.

**Models collected from**

- **RAD-DINO:** Model and weights fetched from huggingface.
- **GLoRIA:** Model collected from GitHub, weights from stanfordmedicine.app.box.com (ResNet-50).
- **BioViL:** Model and weights downloaded through the HI-ML Multimodal Toolbox Python package pypi.org/project/hi-ml-multimodal/.
- **MRM:** Acquired from https://github.com/RL4M/MRM-pytorch.

- **BiomedCLIP:** Model and weights fetched from huggingface.
- **C2L:** ResNet-18 weights downloaded from GitHub.
- **Medical MAE:** ViT-Base/16 weights (0.5M dataset) downloaded from GitHub.

**Model-specific preprocessing**  Image preprocessing was chosen to match the preprocessing each backbone used during initial pre-training closely. All preprocessing not fetched from Huggingface was implemented using torchvision.

- **RAD-DINO:** The preprocessor was fetched from the corresponding huggingface repository.
- **GLoRIA & Medical MAE:** Resized such that the shorter side 238 pixels, followed by a $224 \times 224$ center crop. The pixel values were then rescaled to range $[0, 1]$, and the three channels subsequently normalized according to the ImageNet mean and standard deviation (mean=$[0.485, 0.456, 0.406]$ and std=$[0.229, 0.224, 0.225]$).
- **BioViL:** Images were initially resized such the shorter side was 512 pixels. $448 \times 448$ center-crop was applied followed by rescaling of values to range $[0, 1]$
- **MRM:** Derived from https://github.com/RL4M/MRM-pytorch. Resize to 224 pixels followed by $224 \times 224$ center crop. The image is converted to grayscale, rescaled to range $[0, 1]$ and normalized with mean=0.4978 and std=0.2449.
- **BiomedCLIP:** Fetched from huggingface.
- **C2L:** Derived from GitHub. Image resize to 224 pixels followed by a $224 \times 224$ center-crop. Channels are rescaled to range $[0, 1]$ and normalized according to the ImageNet mean and std (provided previously).

For data augmentation, we used random resized crop between scales 0.4 and 0.9. During training, layer normalization was applied after extracting the pre-trained encoder features (*i.e.*, applied to $V(x)$ and/or $T(z)$). An additional normalization was used for the teacher model before concatenating the (self-attended and mean-pooled) image and text representations. Self-attention layers used a dropout layer ($p = 0.2$) on the attention weights before multiplying them with the value vector.

**Pre-training datasets**  Table 1 offers an overview of the datasets each model has been pre-trained with.

Table 1: An overview of the pre-trained models compared in this study. † indicates that the image and text representations have been explicitly aligned. Size refers to the input image resolution.

| Model ($V(x)$) | Type | Size | Pre-Trained Using | | Pre-Trained On | | |
| --- | --- | --- | --- | --- | --- | --- | --- |
| | | | Image SSL | Text | MIMIC | CheXpert | Other |
| RAD-DINO | ViT-B/14 | 518 | ✓ | ✗ | ✓ | ✓ | ✓ |
| C2L | ResNet-18 | 224 | ✓ | ✗ | ✓ | ✓ | ✓ |
| Medical MAE | ViT-B/16 | 224 | ✓ | ✗ | ✓ | ✓ | ✓ |
| MRM | ViT-B/16 | 224 | ✓ | ✓ | ✓ | ✗ | ✗ |
| BioViL-T | ResNet-50 | 512 | ✗ | ✓† | ✓ | ✗ | ✗ |
| GLoRIA | ResNet-50 | 224 | ✗ | ✓† | ✗ | ✓ | ✗ |
| BiomedCLIP | ViT-B/16 | 224 | ✗ | ✓† | ✗ | ✗ | ✓ |

## A.1 Evaluation Datasets

Details of datasets collected from MIMIC-CXR:

- MIMIC 5x1200 - *Diagnostic* (6,000 images): We perform the common task of predicting diagnostic labels extracted from radiological reports. To do this, we construct MIMIC CXR 5x1200 (similar to CheXpert 5x200 in [8]) by choosing a subset of the MIMIC-CXR-JPG subjects that have had exactly one of five labels assigned to them. Following [13], these labels are Atelectasis, Cardiomegaly, Edema, Pleural Effusion, and Consolidation. The final dataset includes 1200 images per label (1000 for training and 200 for evaluation).

- Age - *Auxiliary* (35,242 images): To evaluate model performance on targets rarely discussed in the report, we predict the age of a patient based on their radiograph. We construct a classification problem by dividing the ages into 5 bins, as done in [10]. For each seed, $90\%$ of images are sampled for training, with $10\%$ withheld for validation.

- 3-Day Discharge - *Prognostic* (18,490 images): By linking the images in MIMIC-CXR to the patient records in MIMIC-IV [27], we gain access to admission information for each patient. As a short-term prognostic target, we predict whether a patient will be discharged in the coming 3 days. For each seed, $90\%$ of images are used for training, and $10\%$ withheld for validation.

Details of datasets collected from INSPECT:

- 12-Month PH - *Prognostic* (5,449 images): A binary classification task where 1 indicates that a patient was diagnosed with pulmonary hypertension (PH) within 12 months of an image being taken. The dataset is collected by removing the censored samples and then artificially balancing it by subsampling the number of patients that did not experience the event. For each seed, $75\%$ of images are sampled for training, while $25\%$ are withheld for validation.

- Readmission - *Prognostic* (5,651): A binary classification task, predicting whether a patient will be readmitted in the coming 12 months. Censored patients were removed, and the dataset was artificially balanced similar to 12-Month PH. For each seed, $75\%$ of images are sampled for training, while $25\%$ are withheld for validation.

**Training sizes and epochs**    Tables 2–6 cover the training dataset sizes, and their corresponding number of training epochs, used for the experiments in the main paper. Epochs were chosen such that all models had time to converge. We additionally plan to release the specific train-test splits for each seed.

Table 2: **MIMIC-CXR-JPG**

| Fraction | #samples | Epochs |
|---|---|---|
| 2.5% | 100 | 100 |
| 5% | 200 | 100 |
| 10% | 400 | 100 |
| 25% | 1000 | 50 |
| 50% | 2000 | 50 |
| 100% | 4000 | 50 |

Table 3: **Age**

| Fraction | #samples | Epochs |
|---|---|---|
| 1% | 317 | 150 |
| 2.5% | 792 | 100 |
| 5% | 1585 | 100 |
| 10% | 3171 | 50 |

Table 4: **3-Day Discharge**

| Fraction | #samples | Epochs |
|---|---|---|
| 0.5% | 82 | 100 |
| 1% | 165 | 100 |
| 2.5% | 416 | 100 |
| 5% | 829 | 50 |
| 10% | 1662 | 50 |
| 25% | 4145 | 50 |

Table 5: **Readmission**

| Fraction | #samples | Epochs |
|---|---|---|
| 5% | 211 | 100 |
| 10% | 423 | 100 |
| 17.5% | 741 | 100 |
| 25% | 1059 | 100 |
| 50% | 2119 | 50 |

Table 6: **12-Month PH**

| Fraction | #samples | Epochs |
|---|---|---|
| 5% | 204 | 100 |
| 10% | 408 | 100 |
| 17.5% | 715 | 100 |
| 25% | 1022 | 100 |
| 50% | 2044 | 50 |

## B  Additional Figures & Results

### B.1  The diversity of image classification tasks

The vast majority of work in medical image classification has focused on *diagnostic tasks*, where the label $Y$ represents the presence of one or more medical conditions [12, 14, 13]. This ignores that clinical prediction challenges frequently involve *prognostic* targets such as future (*e.g.*, 30-day) mortality [28], hospital readmission [4], and disease forecasting [29]. Critically, the causality and association strength between radiograph $X$, report $Z$, and target variable $Y$ can vary substantially depending on the nature of the task. We illustrate this in Figure 4 and give examples below.

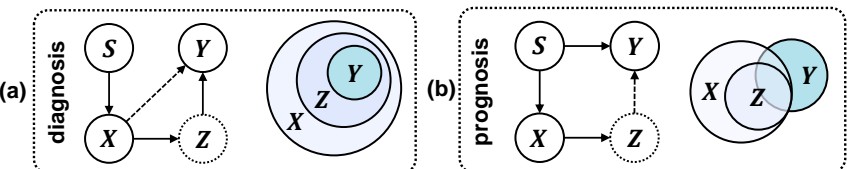

Figure 4: Example causal graphs under the (a) diagnosis or (b) prognosis setting, where $S$: (unobserved) patient state, $X$: medical image, $Z$: radiology report (potentially missing at test time) and $Y$: target label. Dashed edges indicate associations likely to be weak. The accompanying Venn diagrams conceptually illustrate the relationship between the information contained in the observed variables. Note that this is an illustration, and these are not the only possible graphs.

In many *diagnostic* tasks, like pneumonia detection [30], the target label $Y$ is determined almost completely by the radiology report $Z$ (see Figure 4a). Often, the report will contain mentions of a likely diagnosis, and if not, it is written to aid physicians in making one. This is taken to its extreme in currently widely-used benchmarks where the label $Y$ has been extracted from $Z$ based on a set of hand-written rules [14, 12, 13]. In other words, $Y$ is deterministically caused by $Z$. In contrast, *prognostic* tasks, such as predicting 3-day discharge or 12-month readmission [31], the target variable $Y$ is often highly stochastic relative to both the image $X$ and report $Z$ (Figure 4b). Outcomes depend largely on latent disease progression and external factors, while radiographs and reports play only a limited, indirect role by informing treatment decisions.

Figure 4 does not represent all categories of medical image classification. Auxiliary tasks like predicting the age of a patient [10] fit in neither the prognostic nor diagnostic category (we label them *auxiliary*), and their causal graphs are harder to determine. Yet, it is clear that the nature of the task affects how much information the report $Z$ carries about $Y$, and whether it can be extracted from $X$. Consequently, *we need richer benchmarks to fully understand the effectiveness of utilizing reports in pre-training and fine-tuning of medical image classifiers*.

### B.2  Performance of Text-Only Models

To gauge how predictive the radiology reports are of the target label (i.e., how well we can predict $Y$ from $Z$), Table 7 compares the AUC of a text-only model and the best-performing image model.

Table 7: A comparison of text-only and image-only models. Text results come from fine-tuning the BioViL-T BERT model, "Image AUC" corresponds to the best-performing model.

| Dataset | M 5x1200 | Age | 3-Day | 12-Month PH | Readmission |
|---|---|---|---|---|---|
| Fraction | 5% | 1% | 1% | 5% | 5% |
| Text AUC (sd) | **96.4** (0.0) | 78.5 (0.9) | 72.5 (1.5) | **71.0** (1.7) | **57.4** (1.9) |
| Image AUC (sd) | 79.9 (1.1) | **92.1** (0.3) | **77.4** (0.9) | 68.5 (2.1) | 56.7 (1.4) |
| Image Model | MRM | Medical MAE | BioViL | MRM | Medical MAE |
| Fraction | 50% | 10% | 10% | 50% | 50% |
| Text AUC (sd) | **98.5** (0.0) | 80.6 (0.2) | 76.0 (0.8) | **76.6** (0.8) | 60.7 (1.4) |
| Image AUC (sd) | 84.2 (0.2) | **94.8** (0.0) | **79.4** (0.9) | 75.2 (1.2) | **61.3** (1.5) |
| Image Model | MRM | Medical MAE | MRM | RAD-DINO | RAD-DINO |

## B.3 AUPRC Figures

 Figure 5 shows the AUPRC for the experiments in Figure 2. Micro-averaging was used for the multi-class tasks. We provide the AUPRC for the distillation experiments in Figure 6.

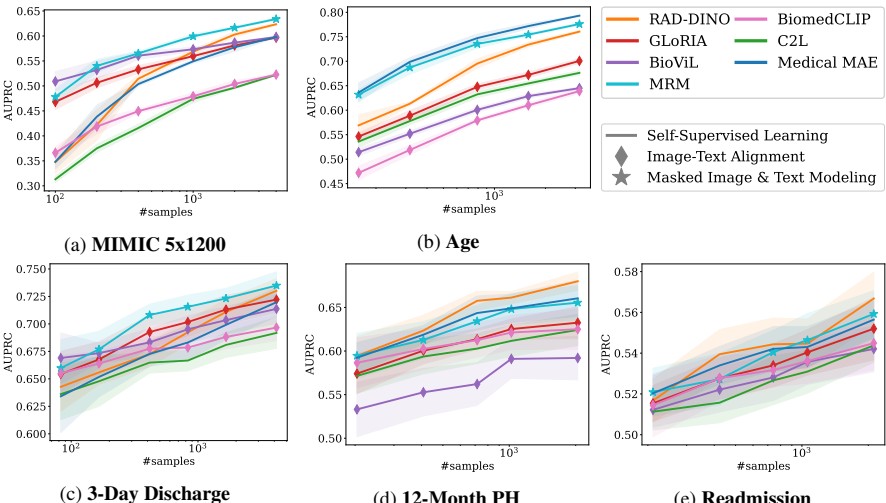

Figure 5: A comparison of the sample efficiency of different backbones. The plots are averaged over 5 seeds, except for Readmission, which is averaged over 10. The shaded area regions represent the 90% CI.

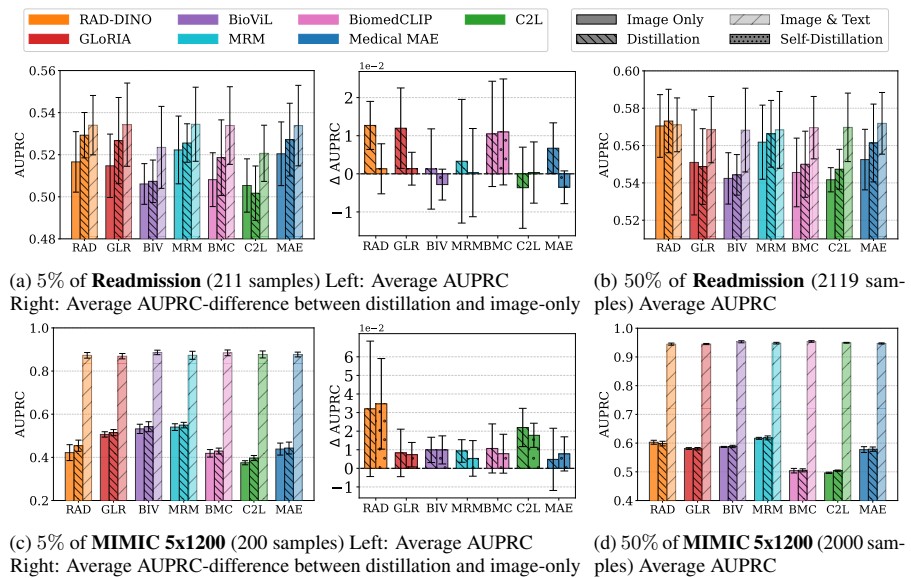

Figure 6: Distillation results on **Readmission** (prognostic) and **MIMIC 5x1200** (diagnostic) with different training set sizes, averaged over 5 seeds. Error bars represent the 95% confidence intervals.

## B.4 Diagnostic Label INSPECT

 We perform an additional experiment on the INSPECT dataset, in which models are trained to predict
 whether or not a patient currently suffers from **Pulmonary Embolism**. As before, this diagnostic
 label has been extracted from the accompanying radiology reports. Similar to **Readmission** and
 **12-Month PH**, we artificially balance the dataset by sub-sampling the number of negative labels to
 match the number of positive. The final dataset consists of $9,375$ images, where $75\%$ are designated

339 for training and 25% for validation. The results (Figure 7) again indicate poor distillation performance
340 when the text is too predictive of the label.

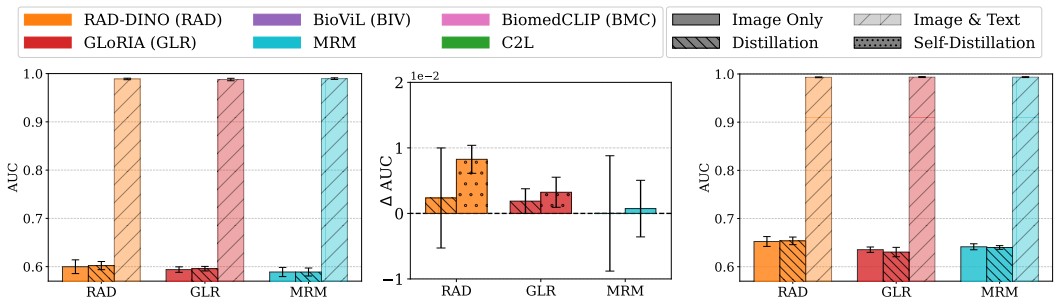

(a) 5% of training data (351 samples) Left: Average AUC
Right: Average AUC-difference between distillation and image-only

(b) 50% of training data (3515 samples) Average AUC

Figure 7: **Pulmonary Embolism**

## B.5 Age and 12-Month PH results

342 We include the distillation results on the **12-Month PH** (Figure 8) and **Age** (Figure 9) datasets.
343 Notably, in the Age experiment, the teacher performs worse on average for every backbone (even
344 with more than $3,000$ training samples), suggesting that the image is substantially more predictive
345 than the radiology reports.

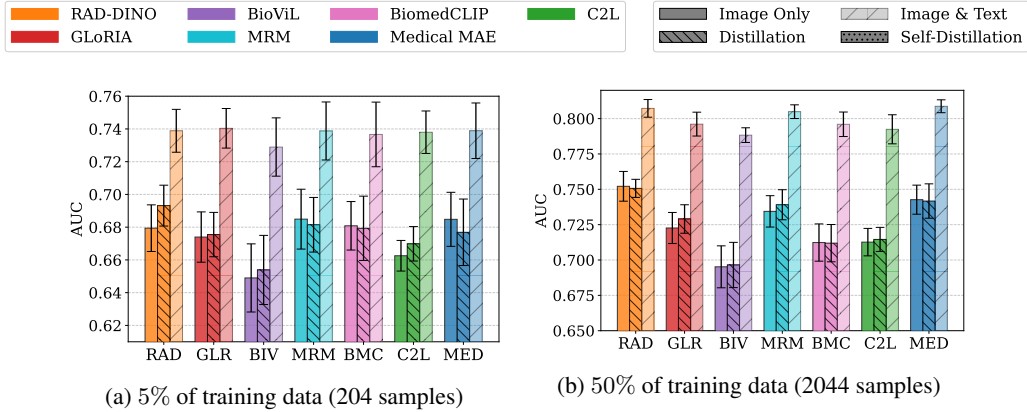

(a) 5% of training data (204 samples)

(b) 50% of training data (2044 samples)

Figure 8: Mean AUC, averaged over 5 seeds, when performing distillation on **12-Month PH**.

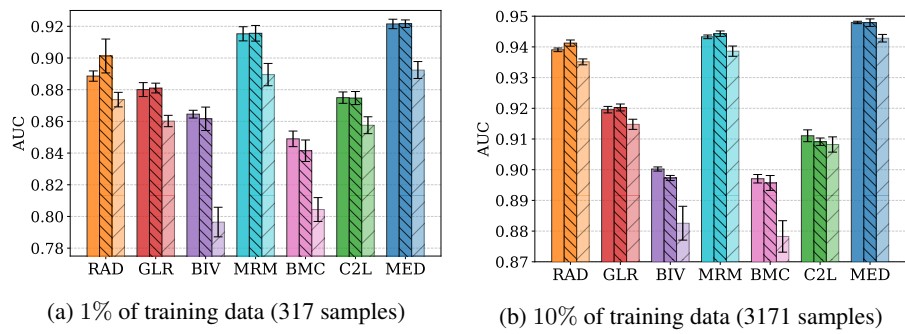

(a) 1% of training data (317 samples)

(b) 10% of training data (3171 samples)

Figure 9: Mean AUC, averaged over 5 seeds, when performing distillation on **Age**.

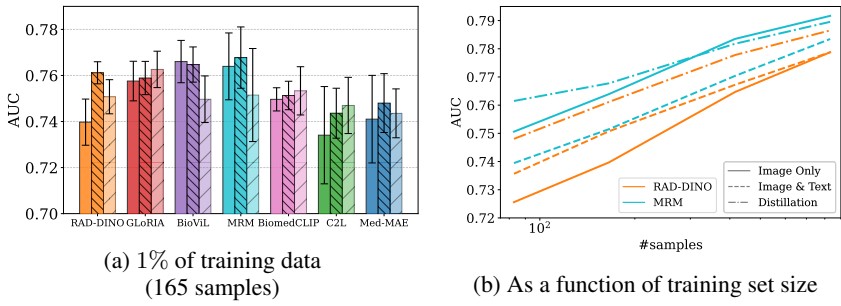

(a) 1% of training data
(165 samples)

(b) As a function of training set size

Figure 10: Mean AUC, averaged over 5 seeds, when performing distillation on **3-Day Discharge**.

## B.6 Dino & MRM Seeds

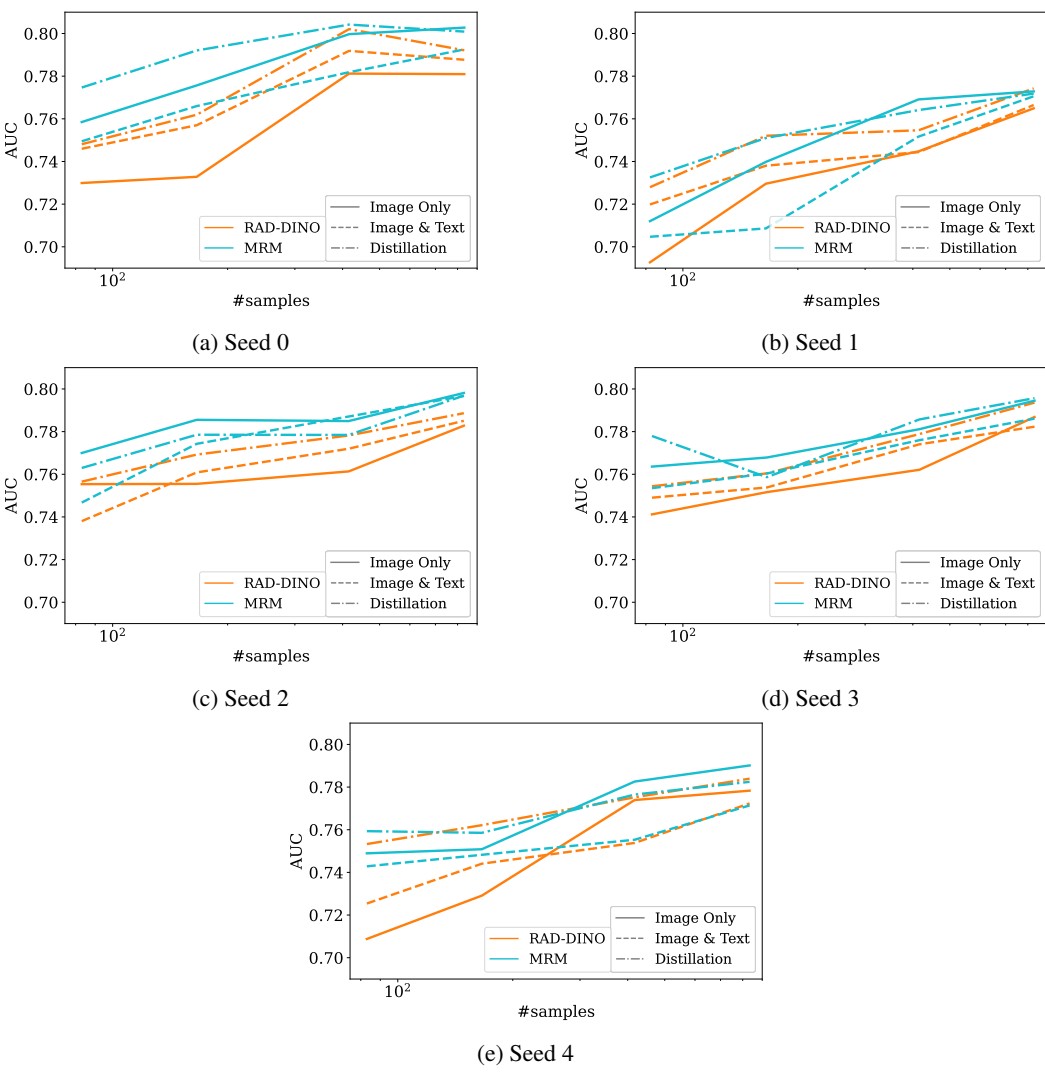

(a) Seed 0

(b) Seed 1

(c) Seed 2

(d) Seed 3

(e) Seed 4

Figure 11: **3-Day Discharge** - RAD-DINO and MRM performance across the 5 seeds averaged in Figure 10b. In the case of RAD-DINO, the student consistently outperforms the teacher.

 **B.7   Readmission and MIMIC 5x1200 Seeds**

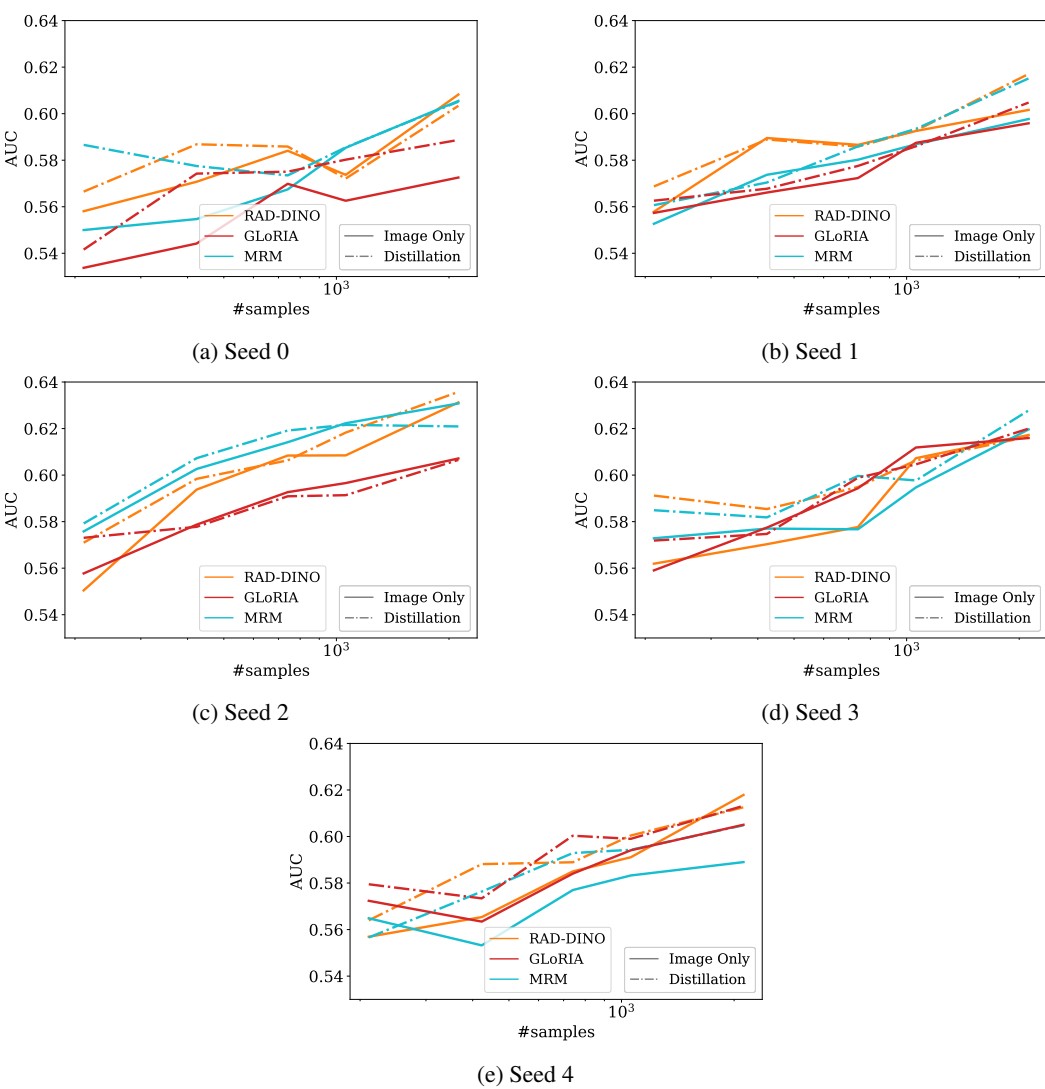

Figure 12: **Readmission** - A comparison of image-only models trained with and without distillation across the 5 seeds averaged in Figures 3a and 3b.

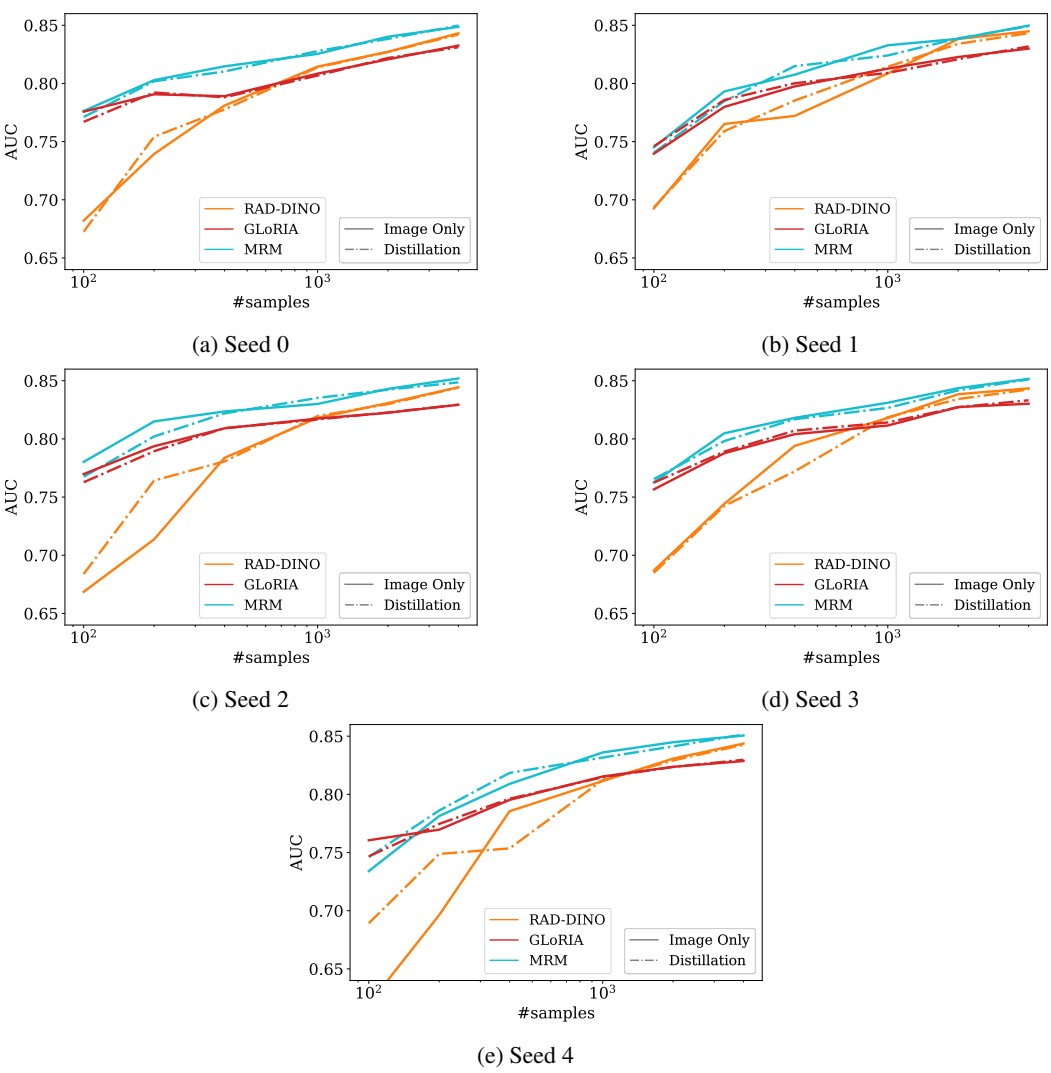

(a) Seed 0

(b) Seed 1

(c) Seed 2

(d) Seed 3

(e) Seed 4

Figure 13: **MIMIC 5x1200** - A comparison of image-only models trained with and without distillation across the 5 seeds averaged in Figures 3c and 3d.

# C    Ablation experiments

## C.1    Impact of Attention Head

Figures 14 and 15 demonstrate the impact of the fine-tuning head. "Attention Head" corresponds to the self-attention head used in all experiments in the main paper. For "Mean - LP", we have instead applied mean pooling over all local embeddings followed by linear probing. Lastly, we evaluated the performance using the RAD-DINO CLS token embedding instead of mean-pooling ("CLS - LP"). We used a learning rate of $10^{-3}$ when performing linear probing (mean- and cls-based). In these experiments, the self-attention head performs noticeably better as the number of samples increases, especially in the case of RAD-DINO. Furthermore, Figure 15 highlights the limitations of only using the CLS token.

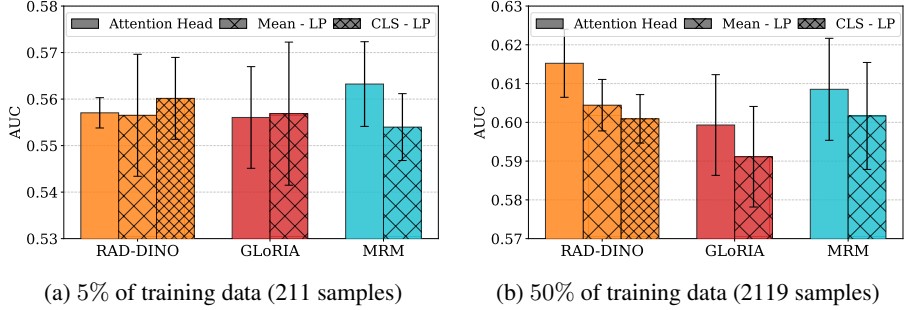

(a) 5% of training data (211 samples)          (b) 50% of training data (2119 samples)

Figure 14: **Readmission** - Ablation of different fine-tuning heads on the image-only model. "Attention Head" is the self-attention head used in all experiments in the main paper, "Mean - LP" is mean-pooling followed by linear probing, and "CLS - LP" uses linear probing on the CLS token embedding.

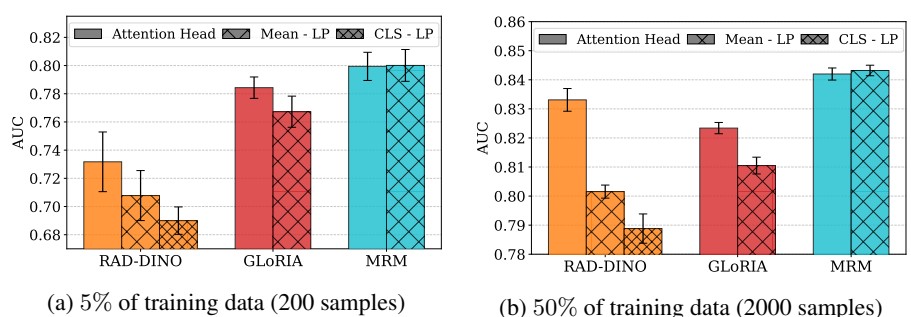

(a) 5% of training data (200 samples)          (b) 50% of training data (2000 samples)

Figure 15: **MIMIC 5x1200** - Ablation of different fine-tuning heads on the image-only model. "Attention Head" is the self-attention head used in all experiments in the main paper, "Mean - LP" is mean-pooling followed by linear probing, and "CLS - LP" uses linear probing on the CLS token embedding.

## C.2    Temperature parameter

We perform an ablation experiment on the temperature parameter ($\tau$) in Equation 1 on **Readmission** (Figure 16) and **MIMIC 5x1200** (Figure 17). The distillation models consistently outperform the image-only baseline, regardless of temperature, on the **Readmission** dataset. On **MIMIC 5x1200**, the choice of $\tau$ seems to have a modest impact (except for RAD-DINO in Figure 17a).

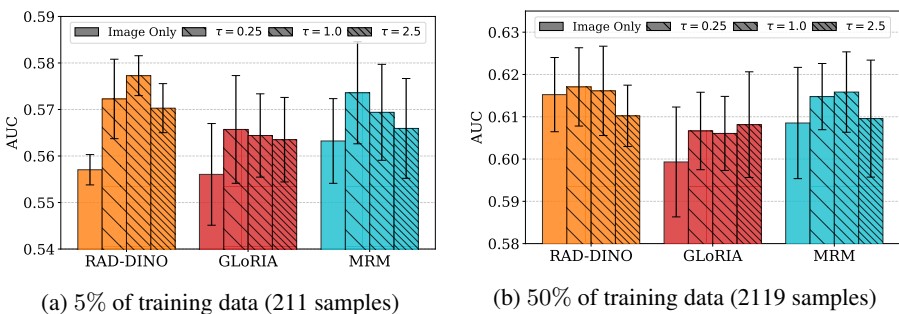

(a) 5% of training data (211 samples)          (b) 50% of training data (2119 samples)

Figure 16: **Readmission** - Distillation performance with different temperatures $\tau$.

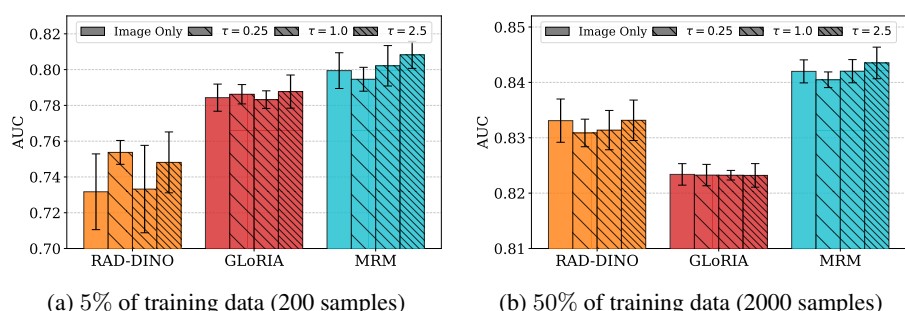

(a) 5% of training data (200 samples)          (b) 50% of training data (2000 samples)

Figure 17: **MIMIC 5x1200** - Distillation performance with different temperatures $\tau$.

## C.3   Choice of text model

To evaluate the impact of the text model in our distillation setup, we train two additional teacher models that each use a different BERT encoder. Apart from the BioViL-T, we use CXR-BERT-general [32], trained with radiology reports, but without the image-based fine-tuning of BioViL-T. We further include Bio_ClinicalBERT [33], which has been trained on clinical notes from MIMIC-III, but not radiology reports. Figures 18 and 19 show the performance of the image-only students distilled from these teachers. The results demonstrate that while the choice of text model impacts distillation quality, benefits are observed for all three on the **Readmission** dataset. While it would be interesting to explore using the accompanying text encoders for the VLM backbones, not all of these are available, and we limit ourselves to the three covered here to make the comparison as fair as possible. CXR-BERT-general and Bio_ClinicalBERT are available on https://huggingface.co/microsoft/BiomedVLP-CXR-BERT-general and https://huggingface.co/emilyalsentzer/Bio_ClinicalBERT respectively.

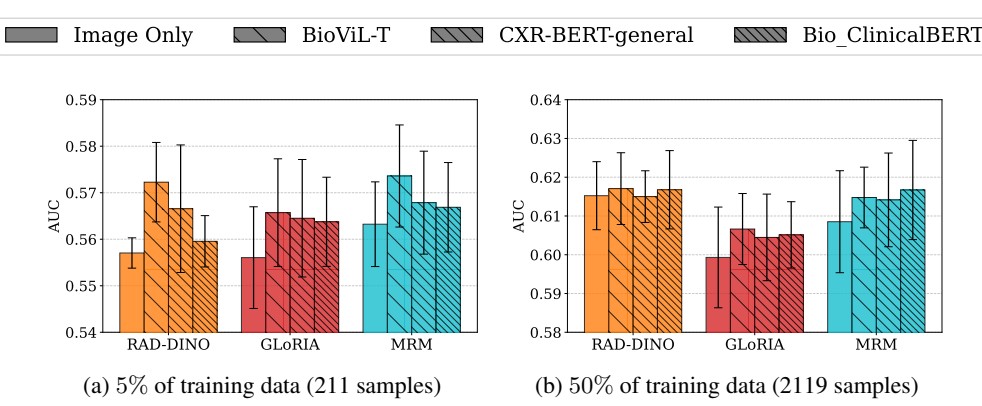

(a) 5% of training data (211 samples)          (b) 50% of training data (2119 samples)

Figure 18: **Readmission** - Distillation from teachers trained with different text backbones.

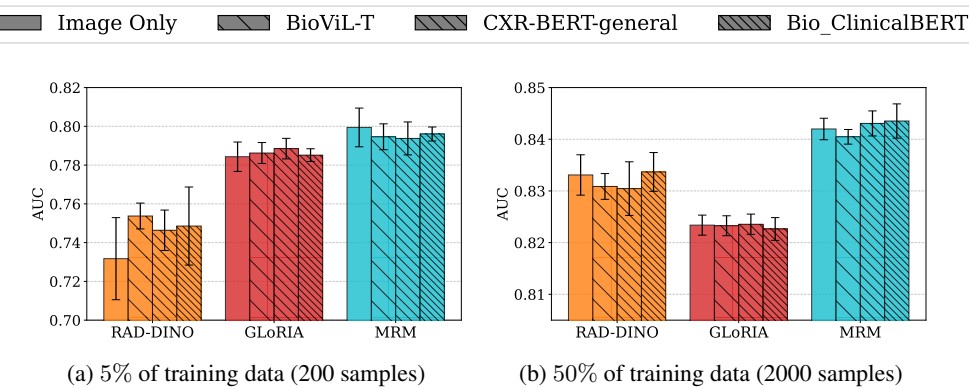

(a) 5% of training data (200 samples)  (b) 50% of training data (2000 samples)

Figure 19: **MIMIC 5x1200** - Distillation from teachers trained with different text backbones.

## D  Compute

Experiments were primarily performed on the NVIDIA T4 (16GB) GPU. RAD-DINO had to be run on an NVIDIA A100 (40GB) to avoid memory constraints. No experiment ran for more than 8 hours, with the majority completing in less than half that time. Given this, we approximate that all experiments run in Figure 2 require less than 1,000 hours. Training of the additional teacher, distillation, and self-distillation models in Figure 3 meant that we had 4 methods for each of the 6 backbones over 5 seeds. Again using the upper limit of 8 hours per run, each subfigure in 3 took less than $8 \cdot 6 \cdot 4 \cdot 5 = 960$ hours to run.

## E  Dataset Licenses

**MIMIC** JPEG images and labels extracted from reports were collected from MIMIC-CXR-JPG 2.1.0 https://physionet.org/content/mimic-cxr-jpg/2.1.0/. Free-text reports and image metadata were fetched from MIMIC-CXR 2.1.0 https://physionet.org/content/mimic-cxr/2.1.0/. Patient admission information (used for the **3-Day Discharge** dataset) was gathered from MIMIC-IV 3.1 https://physionet.org/content/mimiciv/3.1/. All data is provided under the PhysioNet Credentialed Health Data License 1.5.0 https://physionet.org/content/mimic-cxr/view-license/2.1.0/.

**INSPECT** The INSPECT dataset was downloaded from https://stanfordaimi.azurewebsites.net/datasets/318f3464-c4b6-4006-9856-6f48ba40ad67. The data is licensed under the Stanford University Dataset Research Use Agreement, provided on the same page.

