# OpenReview forum: "When are radiology reports useful for training medical image classifiers?"
_EurIPS.cc/2025/Workshop/MedEurIPS — EurIPS 2025 Workshop MedEurIPS Submission_

### Official Review · Reviewer_Lhve · 2025-10-28
**Review comments**

**Rating:** 9
**Confidence:** 5

**Review:**

The paper provides a well-designed, systematic study evaluating how different methods of leveraging radiology reports benefit various training/fine-tuning approaches and downstream tasks.

Suggestions for Improvement:
-Including additional qualitative results showing which parts of the report text are activated would be highly informative.
-I encourage the authors to summarize empirical recommendations for training/fine-tuning recipes based on their findings.
-The potential for future work on training or fine-tuning using generated/synthetic radiology reports is an interesting direction.

Overall, the paper is well-written, well-organized, and easy to follow. The systematic experiments yield many interesting findings that will be highly beneficial for future work in medical Vision-Language Models and broader medical AI model training.

---

### Official Review · Reviewer_3vtr · 2025-10-31
**Review -  When are radiology reports useful for training medical image classifiers?**

**Rating:** 8
**Confidence:** 3

**Review:**

The authors categorize the pretraining of feature encoders into self-supervised, Image-text alignment based and Masked Image and Text Modelling approaches. They aggregate a list of seven commonly used model and assign them to one of the categories.
They evaluate their performance on a set of multiple diagnostic and prognostic tasks.

Advantages:
- Fantastic idea of taking a step back and more generally comparing approach categories to identify broader insights.
- The overall presentation of the paper is of very high quality. The writing is clear, and the experiments were performed over multiple runs to assess reproducibility.
- The experiments and their individual conclusions are nicely presented.

Disadvantages:
- There is no summary or conclusion section, which makes understanding the work and especially the overarching take-home message difficult.
- The experiments seem to assume that the only difference between all methods is the type of pre-training objective, and the results are based on that assumption. However, there are many confounding factors that could also influence the outcomes. It would be beneficial to include ablations focusing only on these components to strengthen the contribution.
- The paper is too condensed to fit into only three pages. Many details are missing to fully understand it. However, since the submissions are non-archival, this is acceptable.


I believe this is a good paper that suffers from the short submission format of MedEURIPS. I am certain that there will be much discussion about this topic and that the paper will be benificial for the workshop. Therefore, I recommend acceptance.

---

### Decision · Program_Chairs · 2025-10-31

**Decision:**

Accept (Oral)

**Comment:**

Both reviewers find the paper exceptionally relevant and well executed. It offers a systematic and thoughtful comparison of how radiology reports contribute to medical image model training, with clear experiments and useful insights.